# Clinical Characteristics and Predictors of Mortality in Critically Ill Influenza Adult Patients

**DOI:** 10.3390/jcm9041073

**Published:** 2020-04-09

**Authors:** Jui-Chi Hsu, Ing-Kit Lee, Wen-Chi Huang, Yi-Chun Chen, Ching-Yen Tsai

**Affiliations:** 1Division of Infectious Diseases, Department of Internal Medicine, Kaohsiung Chang Gung Memorial Hospital, Kaohsiung 833, Taiwan; b9602048@cgmh.org.tw (J.-C.H.); heteyland@cgmh.org.tw (W.-C.H.); sonice83@cgmh.org.tw (Y.-C.C.); greenswallow822@yahoo.com.tw (C.-Y.T.); 2Department of Internal Medicine, Chang Gung University Medical College, Tao-Yuan 330, Taiwan

**Keywords:** severe influenza, pneumonia, acute respiratory distress syndrome, lactate, mortality

## Abstract

Severe influenza is associated with high morbidity and mortality. The aim of this study was to investigate the factors affecting the clinical outcomes of critically ill influenza patients. In this retrospective study, we enrolled critically ill adult patients with influenza at the Kaohsiung Chang Gung Memorial Hospital in Taiwan. We evaluated the demographic, clinical, and laboratory findings and examined whether any of these measurements correlated with mortality. We then created an event-based algorithm as a simple predictive tool using two variables with statistically significant associations with mortality. Between 2015 and 2018, 102 critically ill influenza patients (median age, 62 years) were assessed; among them, 41 (40.1%) patients died. Of the 94 patients who received oseltamivir therapy, 68 (72.3%) began taking oseltamivir 48 h after the onset of illness. Of the 102 patients, the major influenza-associated complications were respiratory failure (97%), pneumonia (94.1%), acute kidney injury (65.7%), adult respiratory distress syndrome (ARDS) (51%), gastrointestinal bleeding (35.3%), and bacteremia (16.7%). In the multivariate regression model, high lactate levels, ARDS, acute kidney injury, and gastrointestinal bleeding were independent predictors of mortality in critically ill influenza patients. The optimal lactate level cutoff for predicting mortality was 3.7 mmol/L with an area under curve of 0.728. We constructed an event-associated algorithm that included lactate and ARDS. Fifteen (75%) of 20 patients with lactate levels 3.7 mmol/L and ARDS died, compared with only 1 (7.7%) of 13 patients with normal lactate levels and without ARDS. We identified clinical and laboratory predictors of mortality that could aid in the care of critically ill influenza patients. Identification of these prognostic markers could be improved to prioritize key examinations that might be useful in determining patient outcomes.

## 1. Introduction

Influenza is an acute viral respiratory infection caused by different types of influenza viruses: influenza A, B, and C [1]. Influenza A subtypes H1N1, H3N2, and influenza B are the most common causes of human influenza [1]. The illness is usually mild and characterized by a sudden onset of fever, cough, sore throat, runny nose, headache, myalgia, and malaise [1]. However, the virus can cause severe illness or even death, especially in high-risk individuals such as young children, the elderly, patients with certain comorbid chronic diseases, and immunocompromised patients [2,3]. Annually, the World Health Organization estimated that global influenza epidemics result in 3 to 5 million cases of severe illness and 290,000 to 650,000 deaths [4,5]. In 2009, a swine-origin influenza A (pandemic 2009 A/H1N1; pdm09 A/H1) emerged and rapidly caused a global pandemic [6]. Remarkably, from 12 April 2009 to 10 April 2010, there were 60.8 million cases and 12,469 deaths in the United States due to pdm09 A/H1 [7]. 

Severe complicated influenza has a significantly high mortality and morbidity [8,9]. Secondary bacterial pneumonia and acute respiratory distress syndrome (ARDS) are some of the common pulmonary complications of influenza, often followed by grave outcomes [10]. In addition to pulmonary complications, extra-pulmonary complications such as meningoencephalitis, myocarditis, and rhabdomyolysis have also been reported to be associated with either influenza A or B [11,12,13]. While early antiviral therapy may reduce complications of influenza [14,15,16], the majority of patients tend to delay seeking medical care and timely diagnosis, leading to the subsequent development of influenza-associated complications, particularly in the elderly and those with comorbid illnesses. Hence, key clinical data are crucial pieces of information that can help clinicians deliver the necessary management to critically ill influenza patients in a timely manner. In the present study, we reviewed the entire clinical course and laboratory data of critically ill patients with influenza and then explored the risk factors independently associated with death as well as constructed a valuable event associated algorithm to help in the assessment of risk of death.

## 2. Methods

### 2.1. Study Design and Patients

We retrospectively studied all critically ill adult patients (aged ≥18 years) with laboratory-confirmed influenza infection admitted between 2015 and 2018, at the Kaohsiung Chang Gung Memorial Hospital, a 2700-bed primary care and tertiary referral medical center that included 5 adult medical and 4 surgical intensive care units (ICUs) in Taiwan. Included critically ill influenza patients were those admitted to an ICU, or those with complications of lower respiratory tract infection and/or multiorgan failure, or those requiring mechanical ventilation during hospitalization. Individuals were excluded from the study if they are <18 years of age; if they are treated as outpatients; or if they had mild flu-like symptoms. Confirmation of influenza virus infection required a positive finding in the respiratory specimen (nasopharyngeal swab and/or pharyngeal swab) by one or more of the following methods: rapid influenza diagnostic test (Formosa One Sure Flu A/B Rapid Test Kit), isolation of the virus in tissue-cell culture (Madin-Darby canine kidney (MDCK) cell line), or reverse-transcriptase–polymerase chain reaction (RT-PCR) (QiAamp Viral RNA Mini Kit; TAIGEN Bioscience Corporation, Taiwan). The choice of diagnostic test (rapid influenza diagnostic test, virus isolation or RT-PCR) for confirming influenza was based on the individual physicians’ judgment.

### 2.2. Data Collection and Definitions

A standardized form for clinical data collection was designed. The data were mainly retrieved from the hospital’s electronic medical records and were supplemented by a secondary manual search. The following data were collected: demographic characteristics, underlying medical conditions, clinical signs and symptoms, antiviral treatment course (oseltamivir or peramivir therapy), results of laboratory tests and radiography findings at the time of presentation and during the entire clinical course, in-hospital complications, and fatality.

Acute respiratory failure was defined as arterial partial pressure of oxygen (PaO_2_) <60 mmHg in ambient air, or tachypnea >30/min. ARDS was defined as acute respiratory distress characterized by bilateral pulmonary consolidation and severe hypoxemia (PaO_2_/Fraction of inspired oxygen ratio <300 mmHg) in the absence of evidence for cardiogenic pulmonary edema [17]. Fulminant hepatitis meant alanine aminotransferase levels (ALT) greater than 16.6 µkat/L. Acute kidney injury was defined as a rapid increase in the serum creatinine level to >44.2 µmol/L compared with that at presentation. Rhabdomyolysis was defined as a five-fold increase in the serum concentrations of creatine phosphokinase above the upper limit of the normal range (reference value, 0.2–2.2 µkat/L), with >95% creatine phosphokinase-muscle fraction. Meningoencephalitis was defined as an altered mental status, fulfilling at least 2 of the following criteria: (1) fever, (2) seizure, (3) focal neurologic signs, (4) abnormality of cerebrospinal fluid, (5) neuroimaging suggestive of encephalitis, and (6) abnormal findings on electroencephalography consistent with encephalitis [18]. A galactomannan cutoff optical density index of >0.5 was used to define positivity for serum and bronchoalveolar lavage samples [19,20]. Mortality was defined as death occurring during the hospital stay for influenza.

### 2.3. Statistical Analysis

To analyze the predictors of mortality among critically ill influenza patients, we initially compared demographic, clinical characteristics, and laboratory findings as well as complications of survivors and nonsurvivors using Fisher’s exact test for categorical variables and Mann-Whitney *U* test for numerical variables. The differences were considered significant at *p* < 0.05. Significant variables in the univariate analyses were entered into a multivariate logistic regression model to identify independent predictors of mortality in critically ill influenza patients. We used receiver operating characteristic curves (ROC) to select cutoff points for independent numerical predictors according to visual assessment of the highest sensitivity and specificity. We then created an event-based algorithm as a simple predictive tool using 2 variables with statistically significant associations with mortality (*p* value less than or equal to 0.001) in the multivariate model. Data were entered and analyzed using the Statistical Package for the Social Sciences statistical software (version 19.0; SPSS Inc., Chicago, IL, USA).

## 3. Results

### 3.1. Patient Characteristics

In total, 102 critically ill patients (97% in medical ICUs and 3% in surgical ICUs) comprising 62 men and 40 women, with a median age of 62 years with laboratory-confirmed influenza virus infection were assessed. The median time from illness onset to hospital presentation was three days (range, 1–14).

Among the 102 patients, three did not receive antiviral therapy (either oseltamivir or peramivir). Of the 99 patients who received antiviral therapy, 89 patients received oseltamivir and five received peramivir; five patients received both oseltamivir and peramivir during their hospitalization. Of the 94 patients who received oseltamivir therapy, 68 (72.3%) began taking oseltamivir 48 h after the onset of illness. Among the 10 patients who received peramivir, 60% received antiviral therapy 48 h after the onset of symptoms. Among 102 patients, influenza A virus was detected in 77 (75.5%) patients (33.3% for pdm09 A/H1, 7.8% for H3N2 and 28.4% for untypable influenza A), influenza B virus in 24 (23.5%), and concurrent influenza A (untypable) and influenza B in 1 (0.9%). The clinical characteristics of the included patients are summarized in Table 1 and Table 2.

### 3.2. Laboratory Testing

The median white blood cell and platelet counts on admission were 7.9 × 10^9^/L and 156.5 × 10^9^/L, respectively. Regarding the laboratory data during hospitalization, the median highest white blood cell and platelet counts and creatinine, ALT, creatine kinase, C-reactive protein, and lactate concentrations were 16.8 × 10^9^/L, 101.5 × 10^9^/L, 183 µmol/L, 1.5 µkat/L (96 patients with data available), 5.5 µkat/L (35 patients with data available), 213.6 mg/L (100 patients with data available), and 3.1 mmol/L (90 patients with available data), respectively. Five (21.7%) of the 23 patients with available data showed serum galactomannan index of >0.5. Six patients underwent bronchoscopy and two had bronchoalveolar lavage fluid galactomannan index of 5.09 and 5.57 with fatal outcomes. *Stenotrophomonas maltophilia* was isolated from bronchoalveolar lavage specimen in one patient. Urinary *Streptococcus* antigen was detected in 3 of the 41 patients with available data. The laboratory characteristics of the included patients are shown in Table 3.

### 3.3. Complications

Table 4 shows the in-hospital complications of the patients during the entire clinical course. The five common complications were respiratory failure (97%), pneumonia (94.1%), acute kidney injury (65.7%), ARDS (51%), and gastrointestinal bleeding (35.3%). Of 59 patients with data available, the median Sequential Organ Failure Assessment score was 6 (range, 2–16). All of the 102 patients were required to be mechanically ventilated, and among them, 90 (88.2%) were ventilated invasively and 12 (11.7%) noninvasively. The median time from illness onset to acute respiratory failure was four days (range, 1–27). Inotropes or vasopressors were used in 44 (43.1%) patients. Ninety-eight (96.1%) patients were admitted to the ICU. Four patients did not admit to ICU; of these, three patients died before ICU admission and the other one improved tachypnea after managed by noninvasive ventilation. 

Extracorporeal membrane oxygenation (ECMO) was initiated for the treatment of refractory hypoxemia, hypercapnia, or both, which occurred despite invasive mechanical ventilation. Seventeen (median age 58 years; range, 24–65) patients with severe influenza-associated ARDS were treated with ECMO, of whom 15 (88.2%) patients had influenza A virus and 2 (11.7%) had influenza B virus infection. The median duration of ECMO support was nine (range, 9–25) days. Of the 17 patients with influenza-associated ARDS who received ECMO treatment, 12 (70.5%) died. In the 12 patients who died, acute kidney injury (*n* = 10), gastrointestinal bleeding (*n* = 8), rhabdomyolysis (*n* = 4), pneumothorax (*n* = 2), intracranial hemorrhage (*n* = 1), and fulminant hepatitis (*n* = 1) were the most common conditions contributing to death. 

Among the 17 bacteremia patients, five (29.4%) patients had bacteremia (*Staphylococcus aureus* in 3, *Pseudomonas aeruginosa* in one, and *Actinomyces oris* and *Streptococcus salivarius* in one) within 48 h after hospitalization. Twelve patients experienced bacteremia after 48 h admission including *Staphylococcus aureus* (*n* = 3), Enterococcus species (*n* = 3), *Stenotrophomonas maltophilia* (*n* = 2), viridans streptococcus (*n* = 1), and *Acinetobacter pittii* (*n* = 1). *Staphylococcus aureus* (35.3%) (16.7% were methicillin resistant) and Enterococcus species (17.6%) were the most frequently isolated bacteria from blood cultures.

### 3.4. Outcomes

Of the 102 patients, 41 (median age, 61 [range, 24–93] years) died, with an overall mortality rate of 40.1%. The median duration of illness before death was 18.5 (range, 2–53) days. Nine patients (21.9%) died within seven days after symptom onset. Of 41 deceased patients, 80.5% had influenza A virus and 19.5% had influenza B virus infection. Of the 38 deceased patients who received oseltamivir treatment, 76.3% of patients received oseltamivir more than 48 h after onset of the illness (Table 4). Among the 41 deceased patients, pneumonia developed in 39 (95.1%) patients, acute kidney injury in 33 (80.5%), ARDS in 27 (65.9%), gastrointestinal bleeding in 21 (51.2%), and bacteremia in 6 (35.3%) patients (Table 4).

### 3.5. Comparison of Survivors and Nonsurvivors

Compared with survivors, nonsurvivors had a significantly shorter length of hospital stay. Upon hospital admission, elevation of myoglobin and creatine kinase–MB isoenzyme levels was significantly associated with mortality. Moreover, a significantly higher white blood cell count and high creatinine, aspartate aminotransferase, C–reactive protein, myoglobin, creatine kinase–MB isoenzyme, and lactate concentrations, in addition to lower platelet count during the course of hospitalization, were reported in nonsurvivors. Nonsurvivors received ECMO treatment and had a significantly higher incidence of acute kidney injury, pneumothorax, ARDS, and gastrointestinal bleeding than survivors. Multivariate analysis showed that high lactate level during hospitalization (adjusted odds ratio [aOR]: 1.041; 95% confidence interval [CI]: 1.019–1.064; *p* < 0.001), ARDS (aOR: 10.098; 95% CI: 2.505–40.714; *p* = 0.001), acute kidney injury (aOR: 9.019; 95% CI: 1.804–45.089; *p* = 0.007), and gastrointestinal bleeding (aOR: 3.828; 95% CI: 1.180–12.415; *p* = 0.025) were independent predictors of mortality in critically ill influenza patients (Table 5).

### 3.6. Event-Based Algorithm

The median highest lactate values (reference value <2.1 mmol/L) and median time from presentation to the highest lactate level among survivors and nonsurvivors were 2.7 mmol/L and 4.3 mmol/L and three days and four days, respectively. We selected the lactate variable in multivariate analyses and plotted the ROC curve to identify the optimal cutoff value for predicting mortality. The optimal cutoff of lactate level for predicting mortality was 3.7 mmol/L with an area under curve of 0.728, and the sensitivity and specificity of this cutoff were estimated at 63.9% and 74.1%, respectively (Figure 1). We then created an event-associated algorithm that included lactate level and ARDS—the two variables that were independently and significantly associated with death in multivariate analyses (Figure 2). Fifteen (75.0%) of the 20 patients with lactate levels of 3.7 mmol/L or above and with ARDS died, compared with only one (7.7%) of 13 patients with lactate levels below 2.1 mmol/L and without ARDS (*p* < 0.001) and 1 (10%) of 10 patients with lactate levels between 2.1 mmol/L and 3.7 mmol/L and without ARDS (*p* < 0.001).

## 4. Discussion

In a study involving 444 adult patients with influenza in hospitals in the United States, the mortality rate was 20.9% [21]. Furthermore, a mortality rate of 20.6% was reported by Francisco et al. in their study of 2059 patients admitted to ICUs for influenza infection [22]. In our study, a mortality rate as high as 40% was found in 102 critically ill adult patients with influenza. However, which variables can predict poor patient outcomes after influenza virus infection remain to be elucidated. In the present study, our dataset included clinical signs and symptoms and laboratory results at presentation and the entire course of hospitalization as well as complications during the clinical course. We determined which demographic, clinical, and laboratory findings were associated with death that could help clinicians deliver timely and sufficient treatment to critically ill influenza patients. Our results underscore that high blood lactate levels during hospitalization, ARDS, acute kidney injury, and gastrointestinal bleeding were independent risk factors of mortality in critically ill influenza patients.

High blood lactate levels indicate tissue hypoxia due to increased lactate generation via anaerobic glycolysis [23]. High blood lactate levels have been correlated with poor outcomes in patients with bacterial sepsis and septic shock [24]. In the present study, high blood lactate levels were found to be significantly and independently associated with fatal outcomes in critically ill influenza patients. In addition, nonsurvivors had a significantly higher prevalence of acute kidney injury and gastrointestinal bleeding and received ECMO treatment in our series. Importantly, acute kidney injury and gastrointestinal bleeding have been shown to be independent risk factors of mortality. We believe that these complications are caused by clinicians’ lack of awareness of early detection of organ hypoperfusion. As patients in early phases of hypoperfusion do not always show obvious clinical signs, blood lactate level may be an important marker for this disorder. Thus, timely recognition of organ hypoperfusion and initiation of effective volume replacement to reverse tissue hypoxia are critical steps in preventing mortality and morbidity. Notably, the median time interval from patient arrival to measurement of highest blood lactate was four days in nonsurvivors in our series. Further, the median time from illness onset to fatality was 18.5 days. This finding indicates that blood lactate levels can be a useful early marker assisting clinicians in predicting the outcomes in critically ill influenza patients. 

Our series here showed gastrointestinal bleeding occurred in one-third of included patients and more than half of them with fatal outcome. Stress ulcer prophylaxis for critically ill patients is not universally practiced in our series. The causes of gastrointestinal bleeding in these patients are multifactorial. Our study underlines that clinicians should be alert to possible gastrointestinal bleeding when caring for a critically ill influenza patient because this complication potentially leads to death if it is not recognized early and treated accordingly.

ARDS is a lethal complication of influenza infection [25]. Ortiz et al. estimated that the incidence of influenza-associated acute respiratory failure was 2.7 events per 100,000 person–years [26]. In a study of 58 patients with ARDS, 28 (48.2%) were due to influenza virus infection, and 32.1% of the patients with influenza-associated ARDS received ECMO treatment [27]. Davies et al. reported that the incidence of pdm09 A/H1-associated ARDS sufficient to warrant consideration of ECMO was estimated at 2.6 cases per million population [28]. ARDS is an independent risk factor for hospital mortality in critically ill influenza patients, and the mortality rate can be as high as 52% [29]. The present study results are consistent with previous findings wherein 97% of critically ill influenza patients developed acute respiratory failure with a median time of four days between illness onset and respiratory failure, and 51% of them subsequently developed ARDS during their clinical course. In addition, approximately one-third of the patients with influenza-associated ARDS required ECMO for profound hypoxemic respiratory failure. Our study highlights that severe oxygenation failure occurred rapidly after hospital admission and that clinicians should not delay delivering appropriate rescue therapies as well as deploying ICU resources to meet this treatment requirement, particularly during the influenza epidemic.

In the present study, we established a simple event-associated algorithm including blood lactate level and ARDS for timely detection of critically ill influenza patients who are at greater risk of mortality. Notably, critically ill influenza patients without ARDS but with a blood lactate concentration of 3.7 mmol/L had an in-hospital mortality of 47.1%, and more importantly, the mortality rate increased to 75% for those with high blood lactate (≥3.7 mmol/L) and those that developed ARDS. In contrast, only 7.7% of critically ill influenza patients without ARDS and with normal blood lactate levels died. Considering the high mortality rate among critically ill influenza patients, this event-based algorithm could aid in the timely decision-making process and provision of prompt intensive care for patients with potentially fatal outcomes, particularly in resource-limited areas; some key laboratory tests such as blood lactate might be of greater value than others when allocating limited healthcare resources.

Previous studies have shown that early administration of an antiviral agent is associated with a shorter duration and reduced severity of illness [14,15,16]. Greater benefits were shown with early treatment initiated within two days after the onset of illness [30,31,32]. In our study, the median time from illness onset to hospital presentation was three days and more than two-thirds of the patients received delayed (48 h after illness onset) antiviral treatment. Although the provision of antiviral therapy between survivors and nonsurvivors did not differ significantly in our series, the importance of early treatment with antivirals in critically ill influenza patients cannot be overemphasized.

In our study, bacteremia was detected in 17 critically ill adult patients. Importantly, five of them acquired bacteremia within 48 h after hospitalization, and in three cases, the infection was caused by *Staphylococcus aureus*. A report of the 2009–2010 influenza pandemic among critically ill children revealed that nearly 5% of the patients had bacteremia within 72 h and *Staphylococcus aureus* was the most frequently isolated bacterium, which contributed to the death rate in the current pandemic [33]. In a study of 32 influenza-positive patients (including pediatric and adult patients), poor outcomes were found among patients who were coinfected with influenza viruses and *Staphylococcus aureus* [34]. Although we were unable to conclude whether or not initiating timely additional antimicrobial treatment in critically ill influenza patients led to better clinical outcomes, our findings and previous reports underscore that *Staphylococcus aureus* remains the most important cause of bacterial coinfection in pediatric and adult influenza patients.

Invasive pulmonary *Aspergillus* as a coinfection in patients with severe influenza has been described [35,36,37]. In a cohort study involving seven ICUs over a period of seven influenza seasons showed that influenza and the use of corticosteroids were independent risk factors for invasive aspergillosis [35]. In the present study, invasive pulmonary aspergillosis was confirmed in two deceased influenza patients with high galactomannan index in bronchoalveolar lavage fluid. This finding emphasizes that clinicians should be aware of the risk of invasive aspergillosis in critically ill influenza patients, particularly immunocompromised patients or those receiving corticosteroids. Further studies are needed to understand the incidence, risk factors, and clinical features of invasive pulmonary aspergillosis in influenza patients.

In 2003, an avian influenza virus of H5N1 subtype was isolated from a smuggled duck in Kinmen Island of Taiwan [38]. However, no locally acquired H5N1 disease in humans had been reported in Taiwan [39]. Thus, none of the patients in this series tested positive for H5N1.

This study has several potential limitations. First, given the retrospective nature of the study, information on vaccination status, including pneumococcus and influenza, and some missing laboratory data such as pneumonia severity index and coagulation profile, were not collected. Second, the study population comprised adult patients; therefore, the results cannot be generalized to pediatric patients. However, the strengths of this study include a detailed description of clinical and laboratory information at presentation and the entire hospitalization course of critically ill patients with influenza. We highlighted the key factors associated with poor outcomes for critically ill influenza patients and established a decision-making algorithm that can take advantage of simple clinical and laboratory evaluations. 

## 5. Conclusions

We identified clinical and laboratory predictors of mortality that could aid in the prediction of the poor outcomes in hospitalized critically ill influenza patients, as timely intensive supportive care might be lifesaving. Medical services and ICUs can be overwhelmed during the peak of influenza epidemics, particularly in point-of-care resource-limited areas. Our findings could substantially assist with allocation of resources in the selection of the main key clinical data in primary care at the initial clinical evaluation of critically ill influenza patients.

## Figures and Tables

**Figure 1 jcm-09-01073-f001:**
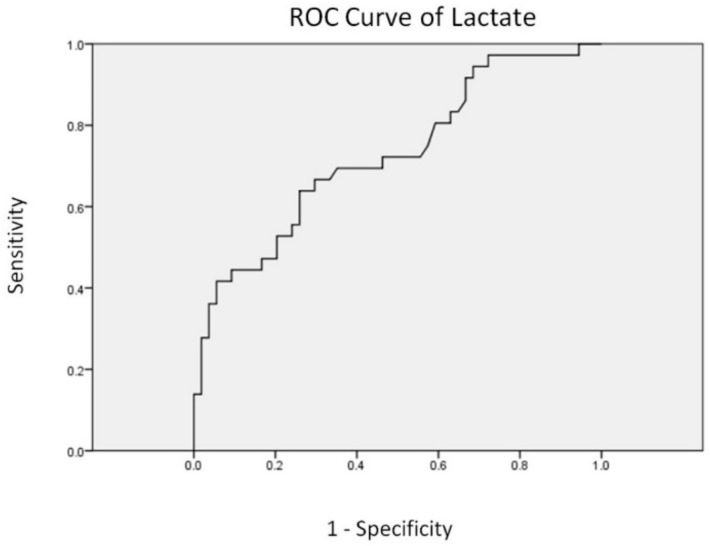
Receiver operating characteristic curve of lactate for prediction of mortality.

**Figure 2 jcm-09-01073-f002:**
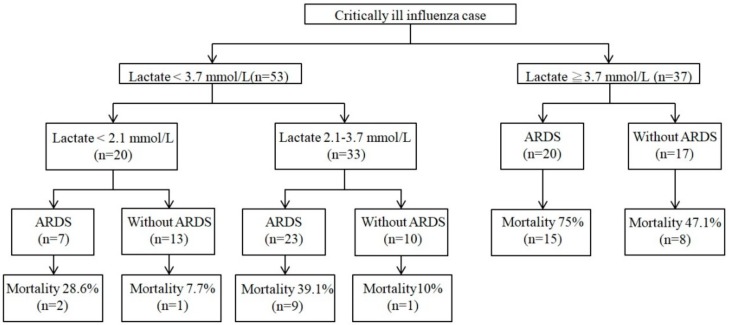
Event-based algorithm.

**Table 1 jcm-09-01073-t001:** Characteristics and diagnostic methods of patients with severe influenza.

	Overall (*n* = 102)	Survivors (*n* = 61)	Nonsurvivors (*n* = 41)	*p* Value
Demographic and Clinical Features				
Age, years, median (range)	62 (24–93)	65 (24–92)	61 (24–93)	0.186
Age group, *N* (%)				0.068
20–49 year	20 (19.6)	11 (18)	9 (22)	
50–64	38 (37.3)	19 (31.1)	19 (46.3)	
≥65 year	44 (43.1)	31 (50.8)	13 (31.7)	
Female gender, *N* (%)	40 (39)	23 (37.7)	17 (41.5)	0.836
Body mass index, median (range)	23.25 (16.7–37.8) (*n* = 86)	23.8 (16.7–35.7) (*n* = 55)	22.7 (17.8–37.8) (*n* = 31)	0.583
**Underlying condition, *N* (%)**				
Bronchial asthma	7 (6.9)	4 (6.6)	3 (7.3)	>0.99
Hypertension	56 (54.9)	38 (62.3)	18 (43.9)	0.073
Type 2 diabetes mellitus	45 (44.1)	28 (45.9)	17 (41.5)	0.689
Chronic kidney disease	16 (15.7)	13 (21.3)	3 (7.3)	0.094
End stage renal disease	9 (8.8)	5 (8.2)	4 (9.8)	>0.99
Chronic obstructive pulmonary disease	8 (7.8)	7 (11.5)	1 (2.4)	0.139
Liver cirrhosis	6 (5.9)	3 (4.9)	3 (7.3)	0.682
qSOFA, median (range)	1 (0–3) (*n* = 99)	1 (0–3) (*n* = 60)	1 (0–2) (*n* = 39)	0.906
SOFA, median (range)	6 (2–16) (*n* = 59)	6 (2–16) (*n* = 34)	7 (3–14) (*n* = 25)	0.263
Times from illness onset to hospital presentation, days, median (range)	3 (1–14)	2 (1–10)	3 (1–14)	0.414
Times from illness onset to fatality, day, median (range)	-	-	18.5 (2–53)	-
Hospital length of stay, days, median (range)	23 (1–107)	32 (2–107)	14 (1–53)	<0.001
Stay in intensive care unit, *N* (%)	98 (96.1)	60 (98.3)	38 (92.7)	0.177
Use antibiotic at presentation, *N* (%)	100 (98)	60 (98.4)	40 (97.6)	>0.99
Use oseltamivir, *N* (%)	94 (92.2)	56 (91.8)	38 (92.7)	>0.99
Use oseltamivir ≥48 h after onset of illness, N/total *N* (%)	68/94 (72.3)	39/56 (69.6)	29/38 (76.3)	0.639
Use peramivir, *N* (%)	10 (9.8)	8 (13.1)	2 (4.9)	0.793
Use peramivir ≥48 h after onset of illness, *N*/total *N* (%)	6/10 (60)	5/8 (62.5)	1/2 (50)	>0.99
Use statin, *N* (%)	21 (20.6)	14 (23)	7 (17.1)	0.610
Use metformin, *N* (%)	17 (16.7)	13 (21.3)	4 (9.8)	0.177
Use vasopressor	44 (43.1)	15 (24.6)	29 (70.7)	<0.001
**Diagnostic methods for influenza, *N*/total *N* (%)**				
Positive of influenza rapid test	51/93 (54.8)	34/56 (60.7)	17/37 (45.9)	-
Positive of RT-PCR for influenza	94/98 (95.9)	59/61 (96.7)	35/37 (94.6)	-
Positive of throat influenza viral culture	37/95 (38.9)	23/59 (39)	14/36 (38.9)	-
**Influenza virus subtype, *N* (%)**				
Influenza A	77 (75.5)	44 (72.1)	33 (80.5)	0.360
pdm 09 H1N1	34 (33.3)	23 (37.7)	11 (26.8)	0.290
H3N2	8 (7.8)	7 (11.4)	1 (2.4)	0.139
Untypable A	29 (28.4)	13 (21.3)	16 (39)	0.073
Influenza B	24 (23.5)	16 (26.2)	8 (19.5)	0.483
Concurrent influenza A ^a^ and B	1 (0.9)	1 (1.6)	0	>0.99

RT-PCR, Reverse transcription-polymerase chain reaction; qSOFA, quick Sequential Organ Failure Assessment. ^a^ Untypable influenza A.

**Table 2 jcm-09-01073-t002:** Symptom/signs of patients with severe influenza.

Symptom/Sign at Presentation	Overall (*n* = 102)	Survivors (*n* = 61)	Nonsurvivors (*n* = 41)	*p* Value
Fever	81 (79.4)	48 (78.7)	33 (80.5)	>0.99
Rhinorrhea	8 (7.8)	3 (4.9)	5 (12.2)	0.262
Cough	85 (83.3)	49 (80.3)	35 (85.4)	0.602
Sore throat	8 (7.8)	5 (8.2)	3 (7.3)	>0.99
Malaise	24 (23.5)	17 (27.9)	7 (17.1)	0.241
Muscle pain	17 (16.7)	10 (16.4)	3 (7.3)	0.233
Headache	5 (4.9)	2 (3.3)	3 (7.3)	0.389
Vomiting/nausea	6 (5.9)	5 (8.2)	1 (2.4)	0.397
Diarrhea	2 (2)	1 (1.6)	1 (2.4)	>0.99
Abdominal pain	2 (2)	1 (1.6)	1 (2.4)	>0.99
Chest pain	8 (7.8)	5 (8.2)	3 (7.3)	>0.99
Skin rash	1 (1)	1 (1.6)	0	>0.99
Altered consciousness	16 (15.7)	11 (18.0)	5 (12.2)	0.581
Seizure	3 (2.9)	2 (3.3)	1 (2.4)	>0.99
Dyspnea	87 (85.3)	51 (83.6)	36 (87.8)	0.776

Data expressed as number (%).

**Table 3 jcm-09-01073-t003:** Laboratory characteristics of patients with severe influenza.

Variable	Overall (*n* = 102)	Survivors (*n* = 61)	Non–survivors (*n* = 41)	*p*
** Laboratory Data at Presentation**				
WBC, (×10^9^/L), median (range)	7.9 (0.5–178)	8.7 (2.1–27.9)	7 (0.5–178)	0.921
Platelet count, (×10^9^/L), median (range)	156.5 (4–641)	156 (60–641)	158 (4–365)	0.407
Hemoglobin, (g/L), median (range)	123 (72–180)	123 (75–180)	123 (72–165)	0.309
Proportion of hematocrit, median (range)	0.36 (0.21–0.51)	0.37 (0.23–0.51)	0.36 (0.21–0.5)	0.785
BUN, (mmol/L), median (range)	7.7 (2.1–39.2)	7.9 (2.1–39.2)	7.5 (2.1–35.3)	0.521
Creatinine, (µmol/L), median (range)	88.4 (43.3–1502.8)	114.9 (44.2–1502.8)	114.9 (53–1105)	0.771
AST, (µkat/L), median (range)	1 (0.2–65.5) (*n* = 98)	0.9 (0.2–40.2) (*n* = 59)	1.1 (0.2–65.5) (*n* = 39)	0.619
ALT, (µkat/L), median (range)	0.5 (0.1–50) (*n* = 96)	0.5 (0.1–50) (*n* = 60)	0.6 (0.1–15.3) (*n* = 36)	0.691
CRP, (mg/L), median (range)	127 (0.4–1377) (*n* = 100)	71.4 (0.4–380) (*n* = 59)	155.1 (1.3–1377)	0.079
Creatine kinase, (µkat/L), median (range)	5.4 (0.2–248.5) (*n* = 35)	5.7 (0.2–198.5) (*n* = 20)	290 (4.9–248.5) (*n* = 15)	0.542
LDH, (µkat/L), median (range)	8.4 (2.9–149.4) (*n* = 28)	6.7 (2.9–149.4) (*n* = 16)	11.2 (4.8–94.6) (*n* = 12)	0.174
Myoglobin, (nmol/L), median (range)	12.7 (0.003–289.5) (*n* = 37)	6 (0.003–289.5) (*n* = 19)	26.3 (5–252.4) (*n* = 18)	0.010
Troponin–I, (µg/L), median (range)	0.06 (0.01–80) (*n* = 95)	0.06 (0.01–15.9) (*n* = 56)	0.05 (0.01–80) (*n* = 39)	0.447
CK–MB, (µkat/L), median (range)	0.06 (0.006–5.12) (*n* = 84)	2.9 (0.05–1.2) (*n* = 49)	0.1 (0.01–5.1) (*n* = 35)	0.013
Lactate, (mmol/L), median (range)	1.8 (0.7–21.1) (*n* = 90)	1.7 (0.9–14.9) (*n* = 54)	2.0 (0.7–21.1) (*n* = 36)	0.840
**Laboratory Data during Hospitalization**				
Highest WBC, (×10^9^/L), median (range)	16.8 (1.9–187.6)	15.8 (5.2–36.1)	19.6 (1.9–187.6)	0.037
Nadir platelet count, (×10^9^/L), median (range)	101.5 (4–311)	113 (16–311)	68 (4–271)	0.005
Highest BUN, (mmol/L), median (range)	19.8 (2.1–94)	14.6 (3.9–94)	25.3 (2.1–80.7)	0.055
Highest creatinine, (µmol/L), median (range)	183 (0.58–44.2)	122 (45.7–1418.2)	289.7 (45.7–930.2)	0.016
Highest AST, (µkat/L), median (range)	1.8 (0.3–258.8) (*n* = 98)	1.6 (0.3–249) (*n* = 59)	2.6 (0.3–258.8) (*n* = 39)	0.033
Highest ALT, (µkat/L), median (range)	1.5 (0.2–78.2) (*n* = 96)	1.5 (0.2–78.2) (*n* = 60)	1.6 (0.3–65.9) (*n* = 36)	0.226
Highest CRP, (mg/L), median (range)	213.6 (2–2641) (*n* = 100)	186 (2–423.3) (*n* = 59)	258.6 (5.97–2641) (*n* = 41)	0.019
Highest creatine kinase, (µkat/L), median (range)	5.5 (0.2–248.5) (*n* = 35)	5.5 (0.2–199) (*n* = 20)	5.0 (0.7–248.5) (*n* = 15)	0.657
Highest LDH, (µkat/L), median (range)	9.8 (3–149.4) (*n* = 28)	6.7 (2.9–149.4) (*n* = 16)	17.5 (4.8–9.5) (*n* = 12)	0.059
Highest myoglobin, (nmol/L), median (range)	30.3 (0.003–8438.8) (*n* = 37)	6.1 (0.003–396.1) (*n* = 19)	86.5 (5.5–8434.8) (*n* = 18)	0.002
Highest Troponin I, (µg/L), median (range)	0.2 (0.01–80) (*n* = 95)	0.1 (0.01–45.6) (*n* = 56)	0.2 (0.01–80) (*n* = 39)	0.736
Highest CK-MB, (µkat/L), median (range)	0.09 (0.006–13.4) (*n* = 84)	0.07 (0.006–2.3) (*n* = 49)	0.2 (0.003–13.4) (*n* = 35)	0.007
Highest lactate, (mmol/L), median (range)	3.1 (0.9–22.5) (*n* = 90)	2.7 (0.9–15) (*n* = 54)	4.3 (1.0–22.5) (*n* = 36)	<0.001
The time interval from presentation to measurement of highest lactate, day, median (range)	4 (1–51) (*n* = 90)	3 (1–51) (*n* = 54)	4 (1–30) (*n* = 36)	0.606
Positive of BAL galactomannan test, *N*/total *N* (%) (reference >0.5 index)	2/6 (33.3)	0/1 (0)	2/5 (40)	>0.99
Positive of serum galactomannan test, *N*/total *N* (%) (reference >0.5 index)	5/23 (21.7)	2/11 (18.2)	3/12 (25)	>0.99
Positive of urine legionella Ag, *N*/total *N* (%)	0/60	0/36	0/24	–
Positive of urine streptococcus Ag, *N*/total *N* (%)	3/41 (7.3)	3/25 (12)	0/16 (0)	0.268

ALT, Alanine aminotransferase; AST, Aspartate aminotransferase; Ag, Antigen; BAL, Bronchial alveolar lavage; BUN, blood urea nitrogen; CRP, C–reactive protein; CK-MB, Creatine kinase-MB isoenzyme; LDH, Lactate dehydrogenase; WBC, White blood cell.

**Table 4 jcm-09-01073-t004:** In-hospital complications of patients with severe influenza.

Variable	Overall (*n* = 102)	Survivors (*n* = 61)	Nonsurvivors (*n* = 41)	*p*
Acute respiratory failure	99 (97)	58 (95.1)	41 (100)	>0.99
Time from onset illness to respiratory failure, day, median (range)	4 (1–27) (*n* = 99)	3.5 (1–16) (*n* = 58)	4 (1–27)	0.451
Time from hospital presentation to respiratory failure, day, median (range)	1 (1–21) (*n* = 99)	1 (1–13) (*n* = 58)	1 (1–21)	0.921
Use of invasive mechanical ventilator	90 (88.2)	52 (85.2)	38 (92.7)	0.352
Duration of invasive mechanical ventilation, day, median (range)	14 (1–60) (*n* = 90)	17 (3–60) (*n* = 52)	9.5 (1–46) (*n* = 38)	0.042
Acute kidney injury	67 (65.7)	34 (55.7)	33 (80.5)	0.011
Renal replacement therapy	25 (24.5)	11 (18)	14 (34.1)	0.099
Pneumothorax	13 (12.7)	4 (6.6)	9 (22)	0.033
Acute respiratory distress syndrome	52 (51)	25 (41)	27 (65.9)	0.016
Pneumonia	96 (94.1)	57 (93.4)	39 (95.1)	>0.99
Pulmonary edema	3 (2.9)	3 (4.9)	0	0.272
Meningoencephalitis	2 (2)	1 (1.6)	1 (2.4)	>0.99
Intracranial hemorrhage	1 (1)	0	1 (2.4)	0.402
Gastrointestinal bleeding	36 (35.3)	15 (24.6)	21 (51.2)	0.011
Fulminant hepatitis	9 (8.8)	6 (9.8)	3 (7.3)	0.737
Rhabdomyolysis	12 (11.8)	5 (8.2)	7 (17.1)	0.216
Bacteremia	17 (16.7)	11 (64.7)	6 (35.3)	>0.99
Bacteremia onset ≥ 48 h after presentation	12 (11.7)	7 (11.4)	5 (12.1)	>0.99
Fungemia	2 (2)	0	2 (4.9)	0.159
ECMO support	17 (16.7)	5 (8.2)	12 (29.3)	0.017
Duration of ECMO treatment, day, median (range)	9 (3–25) (*n* = 17)	9 (5–25) (*n* = 5)	9.5 (3–25) (*n* = 12)	0.958

Data expressed as number (%) unless otherwise indicated. ECMO, Extra-corporeal membrane oxygenation.

**Table 5 jcm-09-01073-t005:** Multivariate analysis of independent risk factors associated with fatality in patients with critically ill influenza.

	Odds Ratio	95% Confidence Interval	*p*
High blood lactate levels	1.041	1.019–1.064	<0.001
Adult respiratory distress syndrome	10.098	2.505–40.714	0.001
Acute kidney injury	9.019	1.804–45.089	0.007
Gastrointestinal bleeding	3.828	1.180–12.415	0.025

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
