# Peer review of "Clinical Characteristics and Predictors of Mortality in Critically Ill Influenza Adult Patients"

_jcm, 2020, doi:10.3390/jcm9041073_

Round 1
Reviewer 1 Report
In the manuscript, the authors investigated the clinical features based on critically ill influenza adult patients. The 4 independent predictors including lactate were identified from the multivariate regression model. The lactate cutoff value 33 mg/dL was calculated based on the ROC curve.
The major advantage of the study is high originality with many patients included and much first-hand data. Basically speaking, it's a good study. However, there are still some issues that can be improved.
1. Line 115: 94 & 10 patients are not correct here since the patients with both medicines were already indicated. Please include the patients with oseltamivir OR peramivir treatment only. Please revise the following sentences as well.
2. The authors used pdm09 in the text (line 120) but Pmd09 in Table 1. Please be consistent.
3. I'd like to suggest showing the ROC curve and lactate cutoff value to replace the non-statistically significant table.
4. I don't know how many patients were infected by H5N1 as the table did not show all influenza subtypes. The highly virulent subtype H5N1 can cause cytokine storms and more likely cause tissue damage to the young population than those senior people. It's interesting to see if the laboratory and other data are the same as the others.
In short, I think it's a nice study for me after making revisions.
Reviewer 2 Report
Summary: The aim of this single center retrospective study was to determine the predictors of death of critically ill influenza patients at hospital admission. The authors collected medical records from 102 critically ill influenza patients. They highlight as independent predictors of mortality: high lactate levels, ARDS, acute kidney injury, and gastrointestinal bleedind. They constructed an event-associated algorithm that included lactatemia > 33 mg/dL and ARDS (defined by Berlin criteria). They conclude that their algorithm can be used early for patient selection in the event of limited resources.
Broad Comments: This article reports a cohort of patients severely affected by influenza. It highlights the seriousness of this disease with a consequent mortality rate in ICU. The search for early factors of mortality and the implementation of an algorithm is the strong point of this study.
I found this manuscript interesting and easy to read. However, the significance of the study is not very clear to me.
1. The first point concerns the mismatch between the results and the objective and conclusion of the article. The algorithm events and the mortality predictors are not early data. The level of lactate is not that at admission and ARDS, AKI, gastrointestinal bleeding are Complications durin hospitalisation. The end of the introduction, the objective and the conclusion must therefore be completely revised.
2. Despite its monocentric nature, the results of this cohort are interesting and could be used in meta-analyzes. However relevant and standard data are not communicated: BMI, number of patients with mechanical ventilation, length of mechanical ventilation, number of patients with renal replacement therapy, number of patients with vasopressors. During our period of covid 19, it would also be interesting to collect the rate of lymphocytes or other predictors. Surprisingly, there is never used a pneumonia severity score, nor an ICU severity score such as SAPS II or SOFA.
Specific comments:
1. Instructions For Authors of J Clin Med specifies that SI units should be used. For example, lactate level should be communicated with mmol/L.
2. Many results appear both in the text and in the tables. It is possible to reduce the text so that the article is easier to read.
3. Please mention which rapid influenza diagnostic test you are using. In view of your results, its predictive values do not seem efficient. On what criteria do you use it? Why compare mortality in Table 1 according to the sensitivity of the test?
4. Idem for cell cultures.
5. Some of your patients are A but not H1N1 and not H3N2. What strains are circulating in your area ?
6. Mortality under ecmo in your cohort seems significant. Is it in line with literature? Could you study this subgroup of patients? What are your criteria of implantation? What are the causes of death, the duration of ecmo?
7. The definition of critically ill influenza patients is complex. Why did you not take patients admitted to ICU as a definition (96%)? Did the non-admissions to ICU correspond to a saturation of the ICU or to limitations of care? Similarly, the exclusion criterion seems subjective.
8. Your bacteriological data could be completed: bacteremia after 48 h, BAL, PVL+, nosocomial pneumonia…
9. Since your study is monocentric, your organizational methods can be described. Your ICU is specialized in infectious disease, are the patients dispatched to several ICUs, etc.
10. Gastrointestinal bleedind are frequent in your cohort. Please specify the number of cirrhotic patients, your anticoagulation and gastric ulcer prevention procedures in ICU.
Round 2
Reviewer 2 Report
All the points of my comments have been clarified.
I congratulate the authors for their study.